# Task Diversity Shortens the In-Context Learning Plateau

**Jaeyeon Kim**[*]                                                          *jaeyeon_kim@g.harvard.edu*
*Harvard University*

**Sehyun Kwon**[*]                                                          *shyun.kwon@samsung.com*
*Samsung Research*

**Joo Young Choi**                                                          *jychoi@krafton.com*
*KRAFTON*

**Jongho Park**                                                             *jjhpark@berkeley.edu*
*KRAFTON*
*UC Berkeley*

**Jaewoong Cho**                                                            *jwcho@krafton.com*
*KRAFTON*

**Jason D. Lee**                                                            *jasondlee@berkeley.edu*
*UC Berkeley*

**Ernest K. Ryu**                                                           *eryu@math.ucla.edu*
*UCLA*

**Reviewed on OpenReview:** *https://openreview.net/forum?id=7t5DzaJOdB*

## Abstract

In-context learning (ICL) describes a language model's ability to generate outputs based on a set of input demonstrations and a subsequent query. To understand this remarkable capability, researchers have studied simplified, stylized models. These studies have consistently observed long loss plateaus, during which models exhibit minimal improvement, followed by a sudden, rapid surge of learning. In this work, we reveal that training on multiple diverse ICL tasks simultaneously *shortens* the loss plateaus, making each task easier to learn. This finding is surprising as it contradicts the natural intuition that the combined complexity of multiple ICL tasks would lengthen the learning process, not shorten it. Our result suggests that the recent success in large-scale training of language models may be attributed not only to the richness of the data at scale but also to the easier optimization (training) induced by the diversity of natural language training data.

## 1 Introduction

In-context learning (ICL), first reported by Brown et al. (2020) with GPT-3, describes a language model's ability to generate outputs based on a set of input demonstrations and a subsequent query. In ICL, the model discerns the task implied by the context of these demonstrations without explicit descriptions, indicating that the model may internally implement an *algorithm* or engage in a *reasoning process*. To understand this remarkable capability that emerges in language models trained on complex real-world language data, researchers such as Garg et al. (2022) have studied simplified, stylized models. In these studies, transformers are trained from scratch to learn simple functions in context, such as linear regression. We thoroughly review this prior work in Section 1.1.

---

[*]These authors contributed equally as co-first authors.
   Code available at `https://github.com/sehyunkwon/task-diversity-icl`

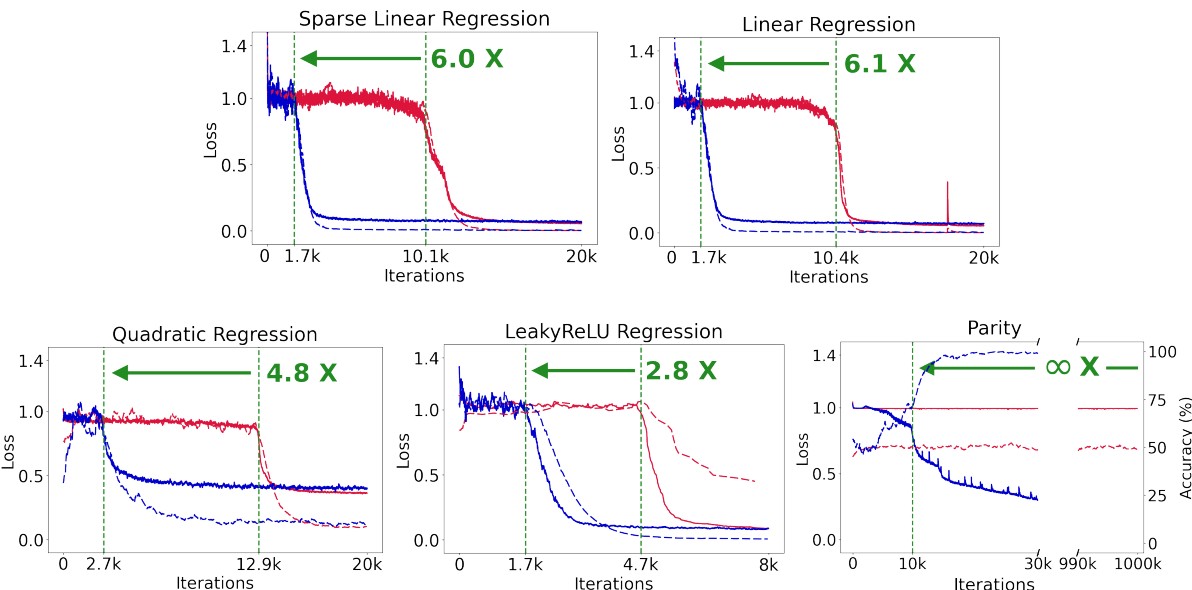

Figure 1: We train a transformer from scratch on in-context learning tasks. Single-task ICL: Training loss (——) and test error/accuracy (- - -) when each task is trained individually. The Parity task cannot be learned even after 1000k training steps. Multi-task ICL: Training loss (——) and test error/accuracy (- - -) when all five tasks are trained simultaneously. Green lines mark the plateau escape points. Surprisingly, multi-task ICL training significantly shortens these plateaus, making training easier.

An intriguing phenomenon observed in these works is the *long loss plateau* in training for ICL. Throughout these plateaus, models display minimal performance improvement, followed by a sudden, rapid surge of learning—deviating from the typical smooth reduction in training loss. In this work, we reveal that **training on multiple ICL tasks simultaneously shortens the loss plateaus** as illustrated in Figure 1. This finding is surprising as it contradicts the natural intuition that the combined complexity of multiple ICL tasks would lengthen the learning process, not shorten it.

In the language of multi-task learning, our findings present an instance where multi-task learning is *easier* than single-task learning in the sense of training dynamics. While previous research has primarily focused on the *statistical* benefits of task diversity in multi-task learning, our findings reveal *optimization* benefits. This insight suggests that the recent success in large-scale training of language models may be attributed not only to the richness of the data at scale but also to the easier optimization (training) induced by the diversity of natural language training data.

**Organization.** Section 2 describes the experimental setup for training for multiple ICL tasks. Section 3 articulates our main claim that task diversity shortens the ICL plateau and presents experimental evidence with transformers and state-space models trained on synthetic and natural language ICL tasks. Section 4 investigates the underlying reasons for this phenomenon. Section 4.1 characterizes the model's behavior during the plateau, which we refer to as the no-context learning regime. Section 4.2 shows that there is a common structure shared across the ICL tasks and that task diversity accelerates the learning of this shared structure. Section 5 presents further experimental details. Section 6 concludes the paper.

## 1.1 Related works

**In-context learning.** In-context learning (ICL) abilities of pretrained Large Language Models have gained significant attention since first investigated by GPT-3 (Brown et al., 2020). Following this, a large body of empirical studies have explored ICL in Large Language Models (Min et al., 2022a;b; Liu et al., 2021; Nie et al., 2022; Rubin et al., 2022; Wei et al., 2023). Given the complexity of real-world data, researchers have

explored ICL in more stylized and simplified setups. Garg et al. (2022) formalized an approach to studying transformer's performance to learn simple function class in context. Building on this work, several works have investigated the ability of models to learn stylized function classes in various settings, including boolean functions (Bhattamishra et al., 2024), regular language (Akyürek et al., 2024), as well as task mixtures (Tripuraneni et al., 2023), and the ability of Mamba (Park et al., 2024; Grazzi et al., 2024; Li et al., 2024). For more comprehensive overview, please refer to the survey by Dong et al. (2024).

While these studies have scrutinized ICL abilities across various setups, most have not delved into the in-context algorithms that are implemented by models. Xie et al. (2022) suggested that the ICL process can be interpreted as Bayesian inference. A number of works have argued, both theoretically and empirically, that transformers implement gradient descent to learn linear regression in-context (Akyürek et al., 2023; von Oswald et al., 2023; Mahankali et al., 2024; Zhang et al., 2024; Ahn et al., 2023). Beyond these works, many have sought to uncover the internal ICL procedure of transformers, in more complex algorithms (Fu et al., 2023; Giannou et al., 2024; Cheng et al., 2024; von Oswald et al., 2024; Lin & Lee, 2024); when handling more intricate tasks (Wang et al., 2024; Guo et al., 2024; Bai et al., 2023; Lin et al., 2024; Wang et al., 2024).

**Abrupt phase transition in in-context learning.** Many studies (Srivastava et al., 2023; Wei et al., 2022; He et al., 2024; Chan et al., 2022; Raventós et al., 2023; Raventos et al., 2023) have shown that a model's ability to perform ICL emerges abruptly, with respect to dataset and model size. While these studies do not focus on the training process, abrupt performance gains during training have also been reported in various works. This transition is typically associated with escaping *loss plateaus*. During these plateaus, no performance gains are observed, but once the plateau is escaped, the model begins to learn in-context abruptly. This phenomenon has been observed in various setups (Garg et al., 2022; Bhattamishra et al., 2024; Park et al., 2024; Li et al., 2024; Kirsch et al., 2024), although these works did not primarily focus on it.

Beyond these, several studies have explored loss plateaus themselves. Fu et al. (2024) suggested that models learn the features of dataset during loss plateaus. Chen et al. (2024) theoretically demonstrated that a plateau occurs during training when a one-layer transformer is trained by linear regression ICL task. Olsson et al. (2022) proposed that the plateau escape and the formation of 'induction head' simultaneously happen. Further work by Reddy (2024); Singh et al. (2024); Song et al. (2024), focused on two-layer transformers, explicitly defining the induction head with transformer parameters and characterizing the internal mechanisms behind its formation. In contrast, Section 4.1 offers a new perspective by providing the explicit form of the model's output during plateaus. Furthermore, in Section 4.2.3, we provide a discrepancy between our finding and induction head.

**Task diversity.** Multi-task training (Caruana, 1997; Baxter, 2000) is a widely used approach for model pretraining. In particular, researchers have identified that *task diversity* is crucial for pre-trained models to outperform in downstream tasks, alongside a large body of research across various domains: supervised learning (Tripuraneni et al., 2020; 2021; Du et al., 2021; Maurer et al., 2016; Crawshaw, 2020; Ruder, 2017), reinforcement learning (Zhang et al., 2023a; Jin et al., 2020; Hu et al., 2021; Yang et al., 2021; Collins et al., 2021; Lu et al., 2022; Cheng et al., 2022), and natural language processing (Zhang et al., 2023a; Zhao et al., 2023; Zhang et al., 2023b; Hu et al., 2020; Song et al., 2020; Zhou et al., 2019; Gunasekar et al., 2023; Sharma et al., 2023). These works mostly focused on statistical benefits, whereas our finding emphasize on the optimization benefits.

## 2 Experimental setup

Our experimental setup follows Garg et al. (2022) and Bhattamishra et al. (2024). Consider a function class $\mathcal{F}$ with domain $\mathcal{X}$. A sequence model $M_\theta$ (transformer or state-space model) is trained to identify $f \in \mathcal{F}$ in context and make a prediction on the subsequent query. The training data consists of sequences of the form $P = (x_1, f(x_1), x_2, f(x_2), \ldots, x_{n-1}, f(x_{n-1}), x_n, f(x_n))$, where $f$ is sampled from a distribution $\mathcal{D}_\mathcal{F}$ and $x_1, \ldots, x_n$ are independently sampled from a distribution $\mathcal{D}_\mathcal{X}$. We train $M_\theta$ for the next token-prediction task: The loss over the sequence $P$ is given by $\frac{1}{n}\sum_{i=1}^{n} \ell(M_\theta(P_i), f(x_i))$, where $\ell(\cdot, \cdot)$ is an appropriate loss function and $P_i = (x_1, f(x_1), x_2, f(x_2), \ldots, x_{i-1}, f(x_{i-1}), x_i)$ is the $i$-th prefix for $i = 1, \ldots, n$. In our experiments, $n = 120$ is a predetermined number shared across all tasks. This procedure is illustrated in Figure 2.

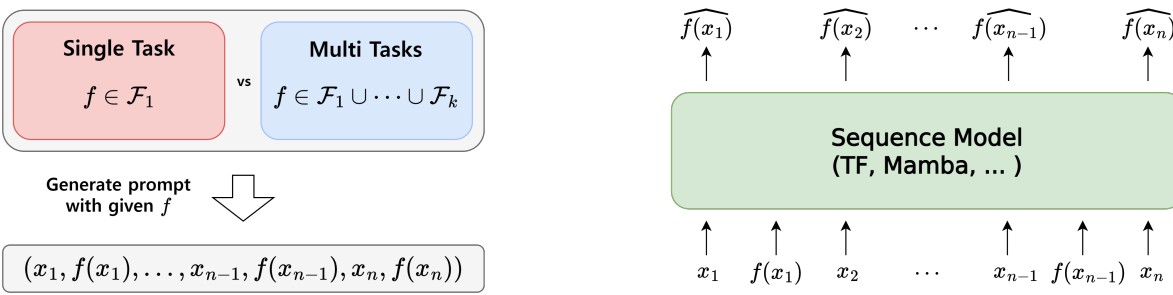

Figure 2: To generate a training sequence, we sample the function $f \in \mathcal{F}_1 \cup \cdots \cup \mathcal{F}_k$, where $\mathcal{F}_1, \ldots, \mathcal{F}_k$ are different in-context function classes, sample $x_1, \ldots, x_n$ IID, and form $(x_1, f(x_1), \ldots, x_{n-1}, f(x_{n-1}), x_n, f(x_n))$. We refer to the case $k = 1$ as single-task ICL and $k > 1$ as multi-task ICL. The sequence model is trained with autoregressive next-token prediction, i.e., the model predicts $f(x_i)$ conditioned on $(x_1, f(x_1), \ldots, x_{i-1}, f(x_{i-1}), x_i)$ for $i = 1, \ldots, n$.

**ICL tasks.** We consider *continuous ICL tasks* and *boolean ICL tasks*. For each ICL task, an $f \in \mathcal{F}$ is chosen, where $\mathcal{F}$ is a function class. For continuous ICL tasks, $\mathcal{F}$ consists of $f \colon \mathbb{R}^d \to \mathbb{R}$ for $d \in \mathbb{N}$ and the probability distribution on the domain is assumed to be $\mathcal{D}_{\mathcal{X}} = \mathcal{N}(0, I_d)$. For boolean ICL tasks, $\mathcal{F}$ consists of $f \colon \{\pm 1\}^d \to \{\pm 1\}$ for $d \in \mathbb{N}$ and the probability distribution on the domain is assumed to be $\mathcal{D}_{\mathcal{X}} = \mathrm{Unif}\left(\{\pm 1\}^d\right)$. We consider 7 different ICL tasks: Linear Regression, Quadratic Regression, Sparse Linear Regression, LeakyReLU Regression, Sparse Parity(2), Sparse Parity(3), Parity. The function class $\mathcal{F}$ defining each of these 7 ICL tasks is precisely stated in Section 5.

**Multi-task ICL training.** Most prior work on ICL, such as Garg et al. (2022); von Oswald et al. (2023); Bhattamishra et al. (2024); Park et al. (2024), considers ICL training with a single function class $\mathcal{F}$. In this work, we train models to learn functions from the union of function classes $\bigcup_{m=1}^{k} \mathcal{F}_m$ in context, where $\mathcal{F}_1, \ldots, \mathcal{F}_k$ are distinct ICL task among the 10 that we list in Section 5. For instance, if the model is trained on the union of linear and quadratic regression tasks, then $\mathcal{F}_1 \cup \mathcal{F}_2 = \{f \mid f(x) = w^\mathsf{T} x\} \cup \{f \mid f(x) = x^\mathsf{T} W x\}$. For each $m = 1, \ldots, k$, we sample $f \sim \mathcal{D}_{\mathcal{F}_m}$ and $x_1, \ldots, x_n \overset{\text{IID}}{\sim} \mathcal{D}_{\mathcal{X}_m}$ and form the sequence $(x_1, f(x_1), x_2, f(x_2), \ldots, x_n, f(x_n))$. This sampling process is repeated $B/k$ times for each $m = 1, \ldots, k$, making $B$ the total batch size. **This choice of batch size ensures a fair comparison between single- and multi-task training, as the number of tokens during training is exactly proportional to the iteration count.** Additionally, we consider a multi-task mixture with uneven probabilities to further support the generality of our claim. To balance the loss scales across different tasks, we normalize each task's loss with constant factors $c_1, \ldots, c_k$ (further discussed in Section 4.1) so that the training loss stabilizes around 1 during the plateaus. Appendix A provides further experimental details. In expectation, we minimize the loss

$$L(\theta) = \sum_{m=1}^{k} c_m \underset{\substack{f \sim \mathcal{D}_{\mathcal{F}_m} \\ x_1, \ldots, x_n \overset{\text{IID}}{\sim} \mathcal{D}_{\mathcal{X}_m}}}{\mathbb{E}} \left[ \frac{1}{n} \sum_{i=1}^{n} \ell\big(M_\theta(P_i), f(x_i)\big) \right].$$

**Test loss.** To evaluate the performance on ICL tasks, we measure the model's prediction error on the last ($n$-th) output of the prompt. For continuous tasks, the test error is $(M_\theta(P_n) - f(x_n))^2$. For boolean tasks, the test accuracy is $\mathbf{1}[\mathrm{sign}(M_\theta(P_n)) = f(x_n)]$, where $\mathbf{1}$ is the indicator function.

## 3 Task diversity shortens ICL plateaus

Long loss plateaus have been commonly reported in the various setups for training sequence models from scratch to perform ICL, including simple in-context functions (Garg et al., 2022; Chen et al., 2024; Li et al., 2024; Bhattamishra et al., 2024; Park et al., 2024), image datasets (Fu et al., 2024; Kirsch et al., 2024; Singh et al., 2024; Reddy, 2024), and language datasets (Akyürek et al., 2024; Olsson et al., 2022). In this section, we present the following claim:

Table 1: **Task diversity shortens the ICL plateau**. We train a transformer with various combinations of 6 different tasks ($d = 10$). For each run, we report two metrics: the time to escape the plateau and the time to complete the learning of the task (written in parentheses). The rows correspond to the number of tasks trained together, and each entry in the table corresponds to the average time across the training runs that include the given task. For example, the entry at (Number of tasks = 3, Sparse Parity(2)) shows the average of $\binom{6}{3}$ results. We find that multi-task training shortens the ICL plateau. To further robustly support our claims, we repeated each experiment three times with varying random seeds and reported the mean values. Details are provided in Appendix C.

| Number of tasks | Boolean Tasks | | Continuous Tasks | | | |
|---|---|---|---|---|---|---|
| | Sparse Parity(2) | Sparse Parity(3) | Linear Regression | Quadratic Regression | LeakyReLU Regression | Sparse Linear Regression |
| 1 | > 1000k | > 1000k | 35.6k (41.1k) | 17.1k (19.2k) | 9.7k (10.7k) | 33.8k (39.6k) |
| 2 | 26.5k (43.9k) | 60.3k (82.6k) | 13.8k (15.8k) | 9.9k (12.1k) | 3.7k (5.0k) | 17.5k (20.5k) |
| 3 | 6.6k (7.1k) | 9.6k (14.9k) | 4.8k (5.8k) | 5.1k (7.2k) | 2.3k (3.0k) | 5.3k (6.4k) |
| 4 | 4.7k (5.5k) | 6.0k (10.5k) | 3.5k (4.5k) | 3.2k (5.8k) | 2.6k (3.4k) | 4.0k (5.0k) |
| 5 | 3.2k (3.9k) | 4.8k (8.0k) | 3.0k (3.9k) | 3.5k (6.0k) | 2.6k (3.5k) | 3.0k (3.9k) |
| 6 | 2.3k (3.0k) | 3.2k (5.0k) | 1.9k (2.4k) | 2.4k (4.8k) | 1.9k (2.4k) | 1.9k (2.4k) |

> **Claim**: Task diversity shortens the ICL plateau, making each task easier to learn.

Here, task diversity refers to learning multi-task ICL on a mixture of distinct function classes. The claim that multi-task ICL is easier to learn than single-task ICL is surprising as it contradicts the natural intuition that the combined complexity of multiple ICL tasks would lengthen the learning process, not shorten it.

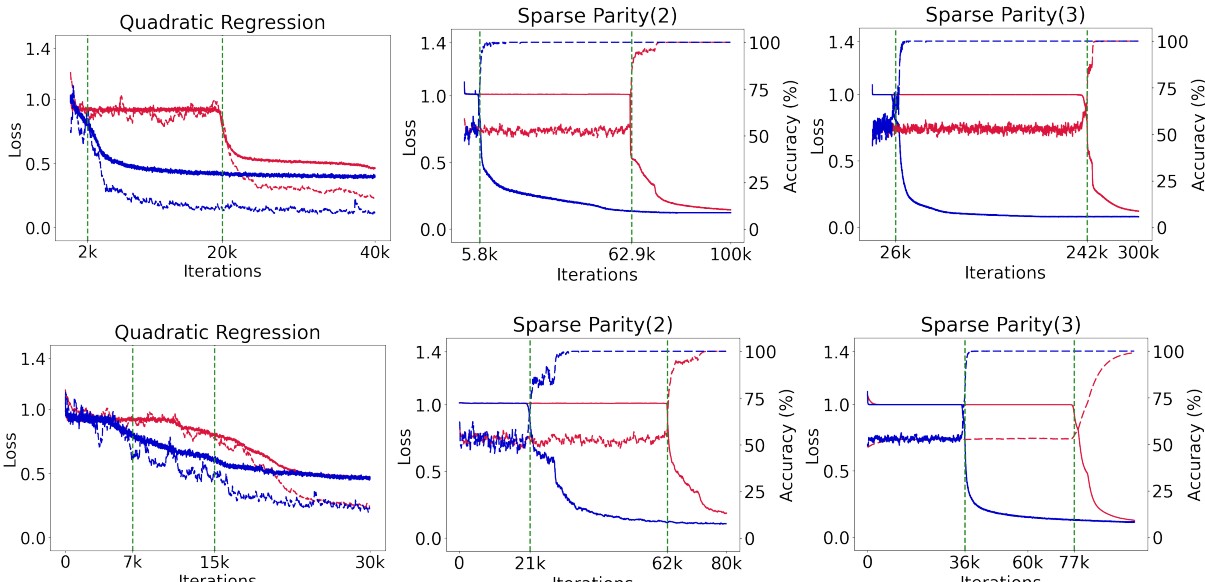

Figure 3: **Mamba (First row)** and **Hyena (Second row)**. Red lines and Blue lines respectively represent the loss dynamics of single-task training and multi-task training. (Left): Quadratic Regression vs. Quadratic Regression+Linear Regression. (Middle): Sparse Parity(2) vs. Sparse Parity(2)+Linear Regression. (Right): Sparse Parity(3) vs. Sparse Parity(3)+Quadratic Regression.

**Generality of our claim.** We provide comprehensive experimental evidence to support this claim. Table 1 summarizes our experimental results on transformers. Figure 3 shows the results of state space models, specifically Mamba (Gu & Dao, 2024) and Hyena (Poli et al., 2023). Across the hundreds of task combinations

Figure 4: Complexity model of multi-task training. The illustration demonstrates the theoretical framework and computational aspects of multi-task training.

in these setups, we consistently observed that task diversity allows training to escape plateaus more quickly. Table 2 presents results for multi-task learning with uneven probabilities of mixture. Figure 7 shows results in natural language ICL tasks supporting our claim. Refer to Appendix A for experimental details.

However, not all ICL tasks mutually reduce the duration of plateaus. For instance, we consider Regbench task (Akyürek et al., 2024), which is generated by a random automata. Our experiments in Appendix D.2 show that combining Regbench with the ICL tasks we consider does not reduce the plateau of Regbench, but does reduce the plateau of the other ICL tasks.

Therefore, the claim should be understood as a description of a general tendency rather than a universal law. Nevertheless, our finding is broadly and robustly observed, as borne out by our extensive experiments.

Table 2: **Muti-task ICL with uneven probabilities.** We conducted experiments on multi-task ICL with uneven mixture probabilities, where 5 ICL tasks are mixed with the ratio $\left(\frac{1}{2}, \frac{1}{8}, \frac{1}{8}, \frac{1}{8}, \frac{1}{8}\right)$. Under these uneven mixtures, we consistently observed a shortened plateau.

|   | SP(2) | SP(3) | LR | QR | Sparse LR |
|---|---|---|---|---|---|
| 1 | $\infty$ | $\infty$ | 2.8k (4.2k) | 10.2k (11.5k) | 2.3k (3.3k) |
| 5 | 2.4k (2.9k) | 3.0k (6.8k) | 2.0k (2.5k) | 2.4k (4.1k) | 2.0k (2.5k) |

**Model of task complexity.** We quickly describe our mental model of the aggregate complexity of multi-task ICL training and the speedup due to task diversity. Let the 'complexity' of an ICL task be the time it takes to escape from the plateau. The complexity of a single-task ICL is the complexity observed when training with just the single task. The complexity of a multi-task ICL is the sum of the complexities of the constituent tasks, but the task diversity reduces it. This reduction makes the aggregate complexity of the multi-task ICL less than the complexity of the individual single-task ICLs. Section 4.2 discusses when and why task diversity could reduce individual complexities. Figure 4 illustrates this notion.

**Escaping plateau ≈ training completion.** In the ICL setups we consider, we observe that (i) there is only one plateau, and (ii) learning is very rapid once this plateau is escaped from, as illustrated in Appendix B. This implies that the time at which training is completed, defined as the moment when the model reaches near-perfect training accuracy, is very close to the time of escaping from the (first) plateau. The results of Table 1 confirm this.

It should be noted that previous studies have demonstrated that multi-stage learning does occur in supervised learning, both theoretically (Ghosh et al., 2022; Bietti et al., 2022; Jin et al., 2023; Wang & Ma, 2023; Berthier et al., 2024) and empirically (Nakkiran et al., 2019; Refinetti et al., 2023; Rubruck et al., 2024). Therefore, we expect that ICL tasks involving complex hierarchical structures may exhibit multiple plateaus.

# 4 Why is task diversity helpful?

In Section 3, we demonstrated that task diversity shortens plateaus but did not explore *why* this effect occurs. In this section, we provide partial answers and hypotheses toward understanding this phenomenon.

## 4.1 Plateau is task-wise no-context learning

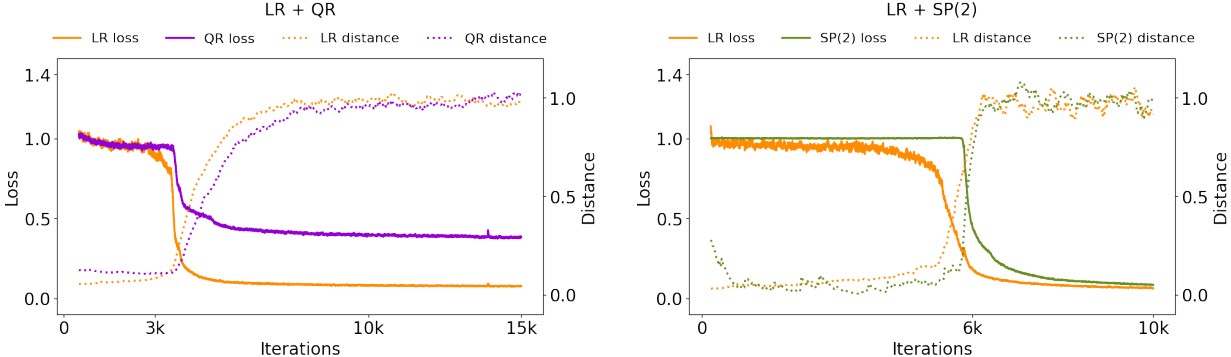

Figure 5: During the plateau, the model output very closely matches the task-wise optimal no-context function. Solid lines (——) denote the raining loss while the dotted lines (- - -) denote the squared distance between model output and task-wise optimal no-context function. **(Left)**: Linear Regression($\mu = -0.5$) + Quadratic Regression($\mu = 0.5$). **(Right)**: Linear Regression($\mu = 1$) + Sparse Parity(2).

At first glance, plateaus might appear to be a failure mode where no meaningful learning occurs. However, in the Sparse Parity(2) task, for example, both test and train accuracies hover around 0.55 during plateaus, as illustrated in Figure 3. If the model learned nothing and were making random predictions, the accuracy should be 0.5. This deviation implies the model is learning something.

For continuous ICL tasks, define the optimal *no-context function* as

$$g_{\mathcal{F}}^{\star} := \mathrm{argmin}_{g} \, \mathbb{E}_{f \sim \mathcal{D}_{\mathcal{F}}, x \sim \mathcal{D}_{\mathcal{X}}} \left[ (g(x) - f(x))^2 \right],$$

i.e., $g_{\mathcal{F}}^{\star}$ is the context-independent function that minimizes the test error. For boolean ICL tasks, $g_{\mathcal{F}}^{\star}$ is analogously defined with the argmax of the test accuracy. In many cases, the optimal no-context function has a simple closed-form expression. For instance, if $\mathcal{F} = \{f \mid f(x) = w^{\mathsf{T}}x\}$ and $\mathcal{D}_{\mathcal{F}}$ is given by $w \sim \mathcal{N}(\mu, I_d)$, then $g_{\mathcal{F}}^{\star}(x) = \mu^{\mathsf{T}}x$. As another example, if $\mathcal{F} = \{f \mid f(x) = x^{\mathsf{T}}Wx\}$ and $\mathcal{D}_{\mathcal{F}}$ is given by $W \sim \mathcal{N}(U, I_{d \times d})$, then $g_{\mathcal{F}}^{\star}(x) = x^{\mathsf{T}}Ux$. The ICL plateau corresponds to *task-wise no-context learning*, which we describe in the following. When we train a model $M_{\theta}$ for ICL with function classes $\mathcal{F}_1, \ldots, \mathcal{F}_k$, the model's output during its plateau corresponds to

$$M_{\theta}(P_n) = g_{\mathcal{F}_m}^{\star}(x_n), \quad P_n \text{ is sampled from } \mathcal{F}_m.$$

In other words, the model identifies the function class $\mathcal{F}_m \in \{\mathcal{F}_1, \ldots, \mathcal{F}_k\}$ and then applies the optimal no-context function corresponding to $\mathcal{F}_m$. The in-context demonstrations $(x_1, f(x_1), \ldots, x_{n-1}, f(x_{n-1}))$ are used to determine the function class $\mathcal{F}_m \in \{\mathcal{F}_1, \ldots, \mathcal{F}_k\}$ but not to determine the specific $f \in \mathcal{F}_m$. This claim can be verified by measuring the error between $M_{\theta}(P_n)$ and $g_{\mathcal{F}}^{\star}$, which we plot in Figure 5. Revisiting the Sparse Parity(2) task, the accuracy during plateau is indeed attributed to $0.55 = \mathbb{E}_{f \sim \mathcal{D}_{\mathcal{F}}, x \sim \mathcal{D}_{\mathcal{X}}}[\mathbf{1}(g_{\mathcal{F}}^{\star}(x) = f(x))]$. Appendix G provides further details on no-context learning regime.

In Section 2, we used the scaling factors $c_1, \ldots, c_k$ to normalize each task's empirical loss. Specifically, we set $c_m = 1/\mathbb{E}_{f \sim \mathcal{D}_{\mathcal{F}}, x \sim \mathcal{D}_{\mathcal{X}}} \left[ \ell(g_{\mathcal{F}_m}^{\star}(x), f(x)) \right]$. Our findings on no-context learning imply that the normalized loss will have a plateau of height 1.

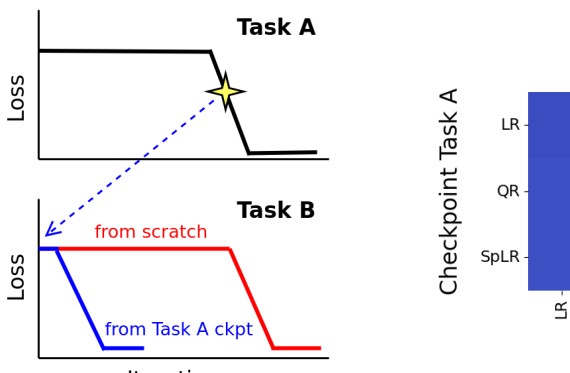

Figure 6: **(Left) Illustration**. We pre-train on Task A and extract a checkpoint ✧ as training escapes the plateau. We then train on Task B starting from the checkpoint. **(Right) Plateau escape time comparison**. Each cell represents the ratio of plateau escape time, with lower ratios (blue color) indicating that the model pre-trained on Task A significantly aids the learning of Task B.

## 4.2 Common structure across ICL tasks

Consider a multi-task ICL setup, where a model is trained on a set of tasks $\bigcup_{m=1}^{k} \mathcal{T}_m$. Suppose there exists a "common structure" $\mathcal{C}$ shared across tasks. Denoting the remaining part of each task as $\mathcal{I}_m$, we can decompose each task as $\mathcal{T}_m = \mathcal{C} + \mathcal{I}_m$ for $m = 1, \dots, k$. Thus, multi-task ICL training is decomposed into two sub-problems: [learning $\mathcal{C}$] and [learning $\mathcal{I}_1, \dots, \mathcal{I}_k$]. We argue that the main claim of Section 3 can be explained by the following hypotheses:

(1) There exists a common structure shared across the multiple ICL tasks.

(2) The ICL plateau arises from the difficulty of learning this common structure.

(3) Training multiple tasks jointly with a shared structure makes it easier to learn that structure.

In the following, we present evidence supporting these hypotheses. Section 4.2.1 demonstrates (1) and (2). Section 4.2.2 elucidates (3) with a toy experiment on feature learning.

### 4.2.1 Checkpoint experiment

Consider the following checkpoint experiment. For each single-task training with task A, we save the model as it escapes the plateau as illustrated in Figure 6 (left). Using this checkpoint model as the initialization, we train it on task B.

The findings, summarized in Figure 6, indicate that the checkpoint model transferred from task A quickly learns task B with a shortened plateau. This implies that (1) task A and task B share a common structure and that (2) the plateau arises from the difficulty of learning the common structure. We conduct an analogous experiment on natural language ICL tasks and observe similar results, as shown in Figure 15 of the appendix. Further details are presented in Appendix E.

### 4.2.2 Generality of our hypothesis

Consider the following intuition. When $\bigcup_{m=1}^{k} \mathcal{T}_m$ are trained concurrently, the model receives multiple "views" of the common structure $\mathcal{C}$ through the different compositions $\mathcal{C} + \mathcal{I}_m$ for $m = 1, \dots, k$. We hypothesize (3): These multiple "views" of $\mathcal{C}$ make $\mathcal{C}$ easier to learn in the sense of a more favorable optimization landscape. This may be the key mechanism allowing task diversity to shorten the ICL plateau, and this phenomenon may extend beyond the ICL setup.

The following feature learning experiment makes this intuition more concrete. Consider the 2-layer feature learning setup, a setup with a large body of prior work (Damian et al., 2022; Ba et al., 2022; Dandi et al., 2023; Wang et al., 2023). For input $x \in \mathbb{R}^d$, the true function to learn is $f_\star(x) = U\sigma(A_\star x)$, where $U \in \mathbb{R}^{k \times h}$ and a true feature matrix $A_\star \in \mathbb{R}^{h \times d}$. The goal is to learn $f \approx f_\star$ with $f(x) = V\sigma(Wx)$, where $h' \gg h$, $V \in \mathbb{R}^{k \times h'}$, and $W \in \mathbb{R}^{h' \times d}$. The loss function is:

$$L(W, V) = \sum_{m=1}^{k} \mathbb{E}_{x \sim \mathcal{D}_x}\left[\left(v_i^\intercal \sigma(Wx) - u_i^\intercal \sigma(A_\star x)\right)^2\right],$$

$$U = \begin{bmatrix} u_1 & \cdots & u_k \end{bmatrix}, V = \begin{bmatrix} v_1 & \cdots & v_k \end{bmatrix}.$$

The idea is that $u_1, \ldots, u_k$ represent $k$ sub-tasks that share the common feature matrix $A_\star$. If we sample $u_1, \ldots, u_k$ from $k$ different distributions, this corresponds to multi-task training of $k$ distinct tasks. Conversely, if we sample $u_1, \ldots, u_k$ from a single distribution, this corresponds to single-task training, as the $k$ sub-tasks become identical. Interestingly, we find that multi-task training exhibits significantly shorter plateau compared to single-task training, as shown in Figure 7 (right). Figure 19 of the appendix shows that additional results with different hyperparameter configurations provide qualitatively similar results.

Although this toy model is a simple supervised learning setup, without sequence models or in-context learning, it reproduces the shortened plateau and makes our intuition more concrete through analogy. The results of this toy model, shown in Figure 7, lead us to make the general hypothesis (3): Training multiple tasks jointly with a shared structure makes it easier to learn the common structure.

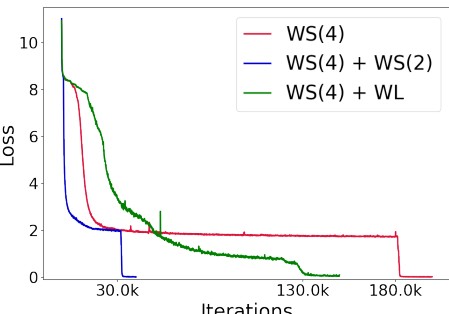 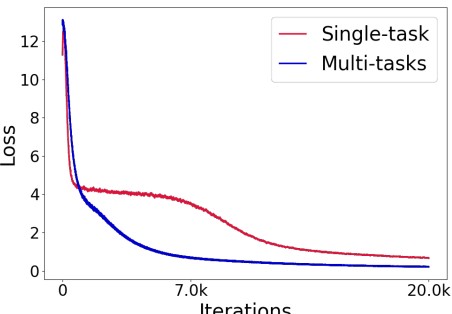

Figure 7: **(Left) Language ICL task**. Previous work Fu et al. (2024) identified the difficulty of learning the WordSelection(4) task. We found that mixing it with WordLength or WordSelection(2) reduces the plateau. Refer to Appendix D.1 for further details. **(Right) Feature learning setup**. For the toy model described in Section 4.2.2, multi-task feature learning significantly shortens the loss plateau. Refer to Appendix F for further details.

### 4.2.3 The common structure is not just an induction head

So then, what specifically is this common structure? We believe it must involve some algorithmic component, as all of the ICL tasks necessitate an internal algorithm to identify the specific function being demonstrated by the in-context demonstrations.

A plausible candidate is the *induction head*, a circuit that searches over the sequence for previous instances of a current token and predicts the same completion again. Indeed, Olsson et al. (2022) argued that the development of an induction head coincides with the escape from the training plateau. To test this idea, we designed the following Retrieval ICL task, inspired by the prior ICL tasks from Park et al. (2024); Singh et al. (2024); Reddy (2024).

**Retrieval.** Sample 1024 5-tuples of one key and four values: $\{(k_i, v_i^1, v_i^2, v_i^3, v_i^4)\}_{i=1}^{1024}$. The $k_i$s and $v_i^j$s are independently sampled from $\mathcal{D}_\mathcal{K}$ and $\mathcal{D}_\mathcal{V}$, respectively. To generate prompts, we uniformly sample $(n-1)$ 5-tuples without replacement and uniformly choose one $v_i$ per 5-tuple, resulting in $\{(\mathbf{k}_i, \mathbf{v}_i)\}_{i=1}^{n-1}$. Next, we sample $p \sim \text{Unif}(\{1, \ldots, n-1\})$ and set $\mathbf{q} = \mathbf{k}_p$. Finally, given $P_n = (\mathbf{k}_1, \mathbf{v}_1, \ldots, \mathbf{k}_{n-1}, \mathbf{v}_{n-1}, \mathbf{q})$, the task is

to predict $\mathbf{v}_p$. We consider two Retrieval tasks: Gaussian Retrieval with $(\mathcal{D}_\mathcal{K}, \mathcal{D}_\mathcal{V}) = (\mathcal{N}(0, I_d), \mathcal{N}(0, 1))$ and Boolean Retrieval with $(\mathcal{D}_\mathcal{K}, \mathcal{D}_\mathcal{V}) = (\mathrm{Unif}\{\pm 1\}^d, \mathrm{Unif}\{\pm 1\})$.

Note that the induction head is precisely the mechanism for solving this Retrieval task. We conduct a checkpoint experiment for the Retrieval tasks, allowing the models to learn the induction head. To train continuous ICL tasks, we use a checkpoint model pre-trained with Gaussian Retrieval. To train Boolean ICL tasks, we use a checkpoint model pre-trained with Boolean Retrieval. We find that the checkpoint does *not* significantly shorten the ICL tasks' plateaus. This result demonstrates that the common structure shared by other ICL tasks is not just an induction head. (It is unclear whether an induction head is useful for our ICL tasks at all.) For further details, please refer to Appendix E.2.

Presently, the problem of characterizing the common structure with any specificity remains unresolved. We defer further investigation of this matter to future work.

## 5 In-context learning tasks

In this section, we quickly list and define the ICL tasks that we consider, which are primarily adapted from Garg et al. (2022); Bhattamishra et al. (2024). Each ICL task is specified by $\mathcal{F}$ the function class, $\mathcal{D}_\mathcal{F}$ a probability distribution over the function class, and $\mathcal{D}_\mathcal{X}$ a probability distribution over the inputs. For continuous ICL tasks, $\mathcal{D}_\mathcal{X} = \mathcal{N}(0, I_d)$. For boolean ICL tasks, $\mathcal{D}_\mathcal{X} = \mathrm{Unif}\{\pm 1\}^d$.

Continuous ICL tasks:

- **Linear Regression.** $\mathcal{F} = \{f \mid f(x) = w^\intercal x\}$. $\mathcal{D}_\mathcal{F}$: Each element of $w \in \mathbb{R}^d$ is independently sampled from $\mathcal{N}(\mu, 1)$.

- **Quadratic Regression.** $\mathcal{F} = \{f \mid f(x) = x^\intercal W x\}$. $\mathcal{D}_\mathcal{F}$: Each element of $W \in \mathbb{R}^{d \times d}$ is independently sampled from $\frac{1}{\sqrt{d}}\mathcal{N}(\mu, 1)$.

- **Sparse Linear Regression.** $\mathcal{F} = \{f \mid f(x) = w_{\mathrm{sparse}}^\intercal x\}$. $\mathcal{D}_\mathcal{F}$: Each element of $w \in \mathbb{R}^d$ is independently sampled from $\mathcal{N}(\mu, 1)$. To sample $w_{\mathrm{sparse}}$, we uniformly choose $k = 3$ coordinates and retain the corresponding coordinates of $w$.

- **LeakyReLU Regression.** $\mathcal{F} = \{f \mid f(x) = \mathrm{LeakyReLU}(w^\intercal x)\}$ with negative slope $\alpha = 0.5$. $\mathcal{D}_\mathcal{F}$: Each elements of $w \in \mathbb{R}^d$ is independently sampled from $\mathcal{N}(\mu, 1)$.

Boolean ICL tasks:

- **Sparse Parity**$(k)$. $\mathcal{F} = \{f \mid f(x) = \prod_{i \in A} x[i]\}$. $\mathcal{D}_\mathcal{F}$: $A \subseteq \{1, \ldots, d\}$ is a uniformly sampled subset of size $k$.

- **Parity.** $\mathcal{F} = \{f \mid f(x) = \prod_{i \in A} x[i]\}$. $\mathcal{D}_\mathcal{F}$: $A \subseteq \{1, \ldots, d\}$ is a uniformly sampled subset, regardless of the size.

## 6 Conclusion

In this work, we identify that training on a diverse set of multiple ICL tasks is surprisingly easier than training for a single ICL task in the sense of a more favorable optimization landscape. This observation aligns with the "blessing of dimensionality/scale" seen in the modern era of deep learning. Indeed, LLM training via next-token prediction can be thought of effectively as a highly diverse multi-task learning, requiring a wide range of reasoning skills for a wide range of text types, and the success of LLM training may be attributed not only to the richness of the data at scale but also to the easier optimization (training) induced by the diversity of natural language training data.

This insight opens new avenues for future work. It may be that explaining and understanding the success of large-scale deep learning requires considering not just the large data, large network, and large compute but also the large (effective) task diversity.

## 7 Acknowledgements

This research was supported by a grant from KRAFTON AI. We thank Wonjong Rhee for initiating this project as part of a course at Seoul National University. We are grateful to Junsu Kim, Jaesung Park, and Hyunsik Chae for their valuable feedback, and to the participants of the KAIST–SNU workshop for their insightful discussions at the early stages of this work. We also thank Chanwoo Park for sharing his empirical insights, and Minkyu Kim for providing feedback on the figures.

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

# A  Experimental details

**Model Architectures.**   We focus on transformer (Vaswani et al., 2017), Mamba (Gu & Dao, 2024), and Hyena (Poli et al., 2023). To handle model architectures that process inputs and outputs as vectors in an embedding space, we first pad each $f(x_i)$ with $(d-1)$ zeros to match the dimension of $x_i$s. We then add a learnable linear layer to map these vectors into the embedding space. A second learnable linear layer maps the model's output to a scalar. Each model's hyperparameters are dictated as follows. For transformer, we use GPT2 (Radford et al., 2019), with 12 decoder layers of embedding size 256 with 8 heads and relative positional encoding. Hyena follows the exact configuration of transformer's setup for the relevant hyperparameters. On the other hand, Mamba employs 24 layers, doubling the number of layers to match the model parameter size, as is standard for Mamba. All models contain input and output projection heads that map input to the hidden state and the hidden state to output. To train a model from scratch, we randomly initialize the model parameters.

**Configuration.**   We use the Adam optimizer (Kingma & Ba, 2015) with a learning rate of 0.0001. We use $B = 64$, making each task's batch size as $64/k$. Regarding the choice of $\ell(\cdot, \cdot)$, we use mean-squared error loss for continuous ICL tasks and cross-entropy loss for boolean ICL tasks. The input dimensions considered are $d = 10$ and 15.

**Plateau and Training Exit Conditions.**   The plateau escape time $t_{\text{plateau}}$ is defined as $t_{\text{plateau}} = \min_{t>100} \left[ \frac{1}{100} \sum_{t'=t-99}^{t} \text{Training loss}(t') < 0.8 \right]$. This effectively captures the plateau escape as we have normalized each task's loss around 1. The training exit conditions are defined as follows $t_{\text{exit}} = \min_{t>100} \left[ \frac{1}{100} \sum_{t'=t-99}^{t} \text{Test error}(t') < 0.2 \right]$ for continuous ICL tasks and $t_{\text{exit}} = \min_{t>100} \left[ \frac{1}{100} \sum_{t'=t-99}^{t} \text{Test accuracy}(t') > 0.95 \right]$ for boolean ICL tasks. $t_{\text{plateau}}$ and $t_{\text{exit}}$ are measured for each individual task.

**Batch generation process**   The following pseudo-algorithm summarizes the batch generation process of (multi)-task ICL training.

After constructing a batch, we minimize the following empirical loss. Denote $f^{(j,m)}$ and $x_i^{(j,m)}$ as the corresponding in-context function and $i$-th input for each $P^{(j,m)}$.

$$\hat{L}(\theta) := \sum_{m=1}^{k} c_m \left[ \sum_{j=1}^{B} \left[ \frac{1}{n} \sum_{i=1}^{n} \ell\big(M_\theta(P_i^{(j,m)}), f^{(j,m)}(x_i^{(j,m)})\big) \right] \right]$$

---

**Algorithm 1** Batch generation process

---

**Require:** Multi-tasks $\{\mathcal{F}_1, \ldots, \mathcal{F}_k\}$
**Require:** Corresponding domains $\{\mathcal{D}_{\mathcal{X}_1}, \ldots, \mathcal{D}_{\mathcal{X}_k}\}$
**Require:** Batch size for each task $\{B_1, \ldots, B_k\}$
**Ensure:** Generated batch of prompts
 1: Initialize Batch $\mathbf{B} \leftarrow \emptyset$
 2: **for** $m = 1$ to $k$ **do**
 3:     **for** $j = 1$ to $B_m$ **do**
 4:         Sample function $f \sim \mathcal{D}_{\mathcal{F}_m}$
 5:         Sample inputs $x_1, \ldots, x_n \overset{\text{IID}}{\sim} \mathcal{D}_{\mathcal{X}_m}$
 6:         Generate prompt: $P^{(j,m)} = (x_1, f(x_1), \ldots, x_{n-1}, f(x_{n-1}), x_n, f(x_n))$
 7:         Update batch $\mathbf{B} \leftarrow \mathbf{B} \cup \{P^{(j,m)}\}$
 8:     **end for**
 9: **end for**

---

## B    Training loss dynamics: One plateau, then rapid drop

In the ICL setups we consider, we observe that (i) there is a single plateau phase, and (ii) learning proceeds very rapidly once the model escapes this plateau. We provide several representative examples of training dynamics that support this observation below. For the cases of Sparse Parity(2) and Sparse Parity(3), training was performed jointly on a combination of three tasks: Sparse Parity(2), Sparse Parity(3), and Sparse Linear Regression.

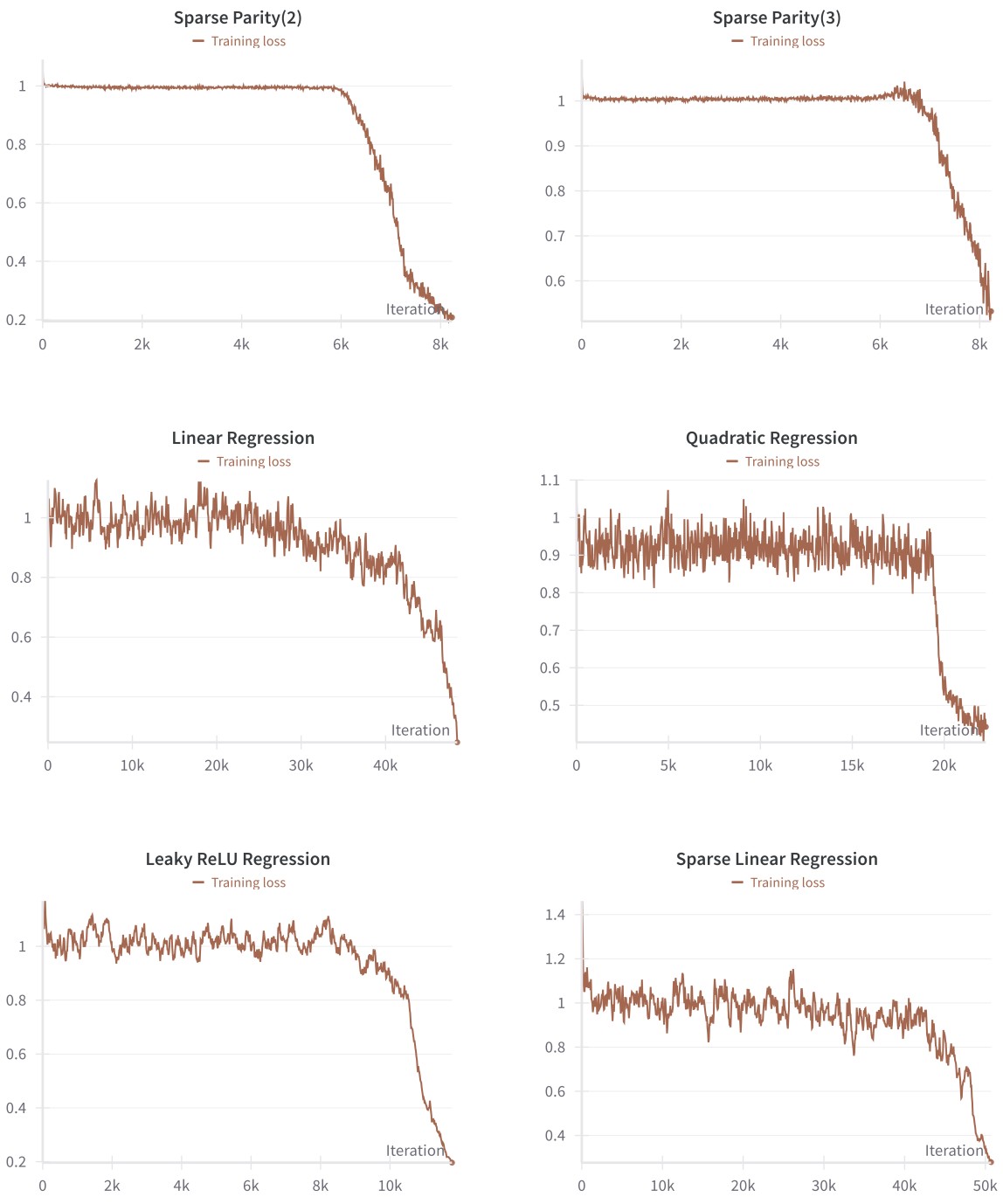

Figure 8: Training loss curves showing a single plateau followed by rapid convergence.

## C   Detailed analysis of table 1

In Table 1, we provided comprehensive experimental evidence demonstrating that task diversity shortens the in-context learning (ICL) plateau, facilitating faster learning. To further robustly support our claims, we repeated each experiment three times with varying random seeds. The figure below plots the mean values along with standard deviations. It clearly shows a strong tendency: as the number of tasks increases, both the time to escape the plateau (blue) and the time to complete learning (orange) decrease significantly.

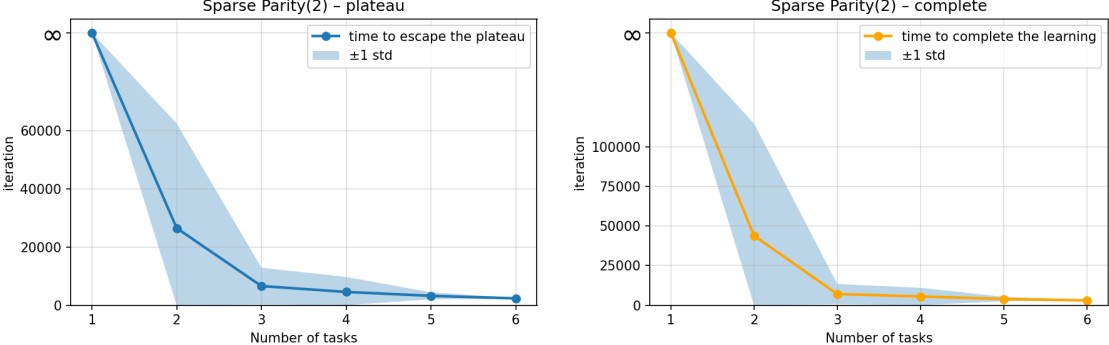

Figure 9: Sparse Parity(2)

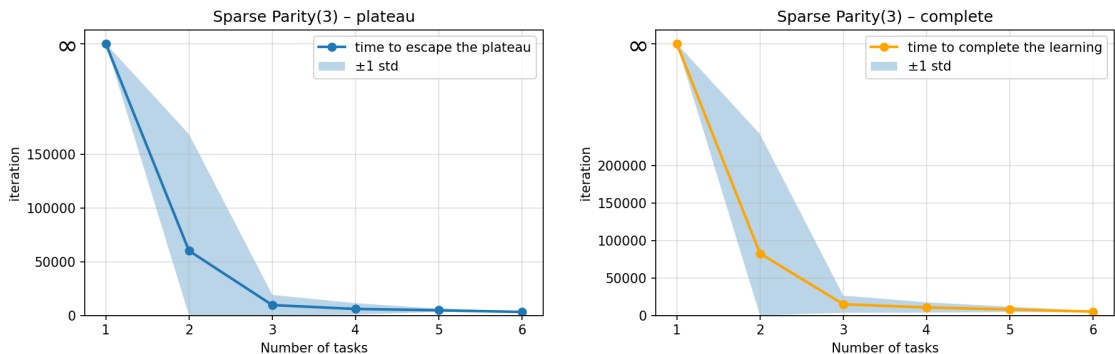

Figure 10: Sparse Parity(3)

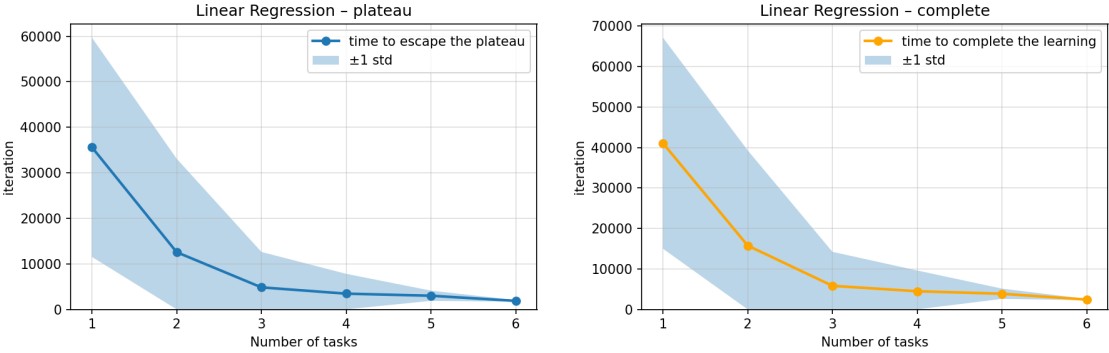

Figure 11: Linear Regression

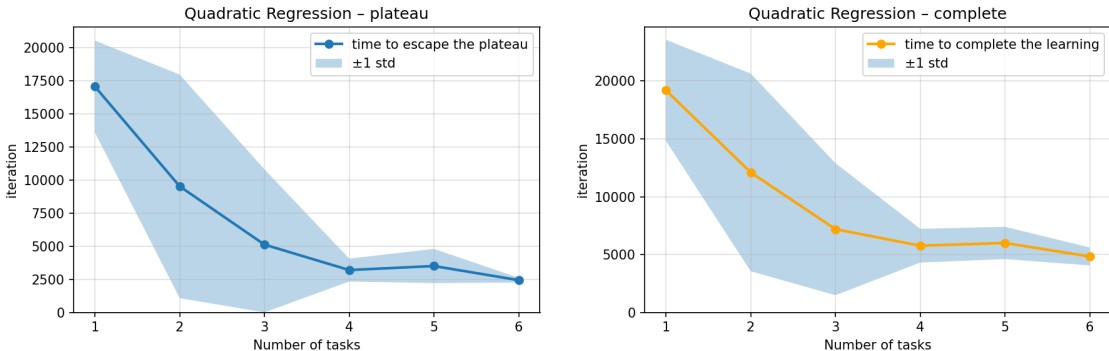

Figure 12: Quadratic Regression

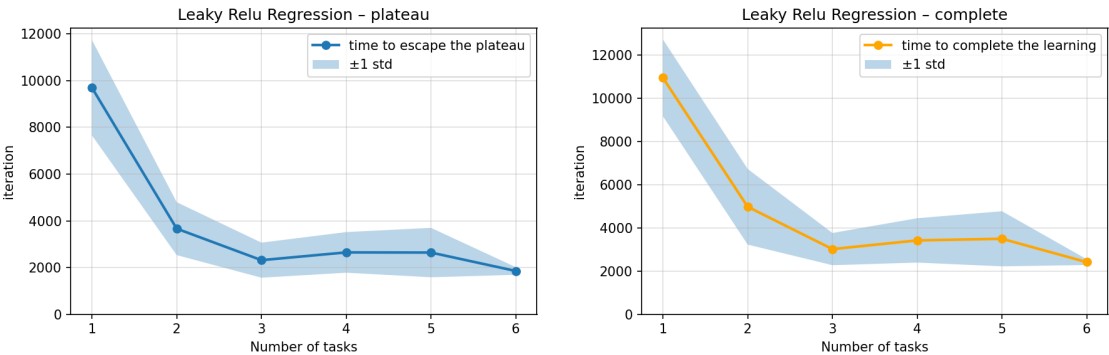

Figure 13: Leaky ReLu Regression

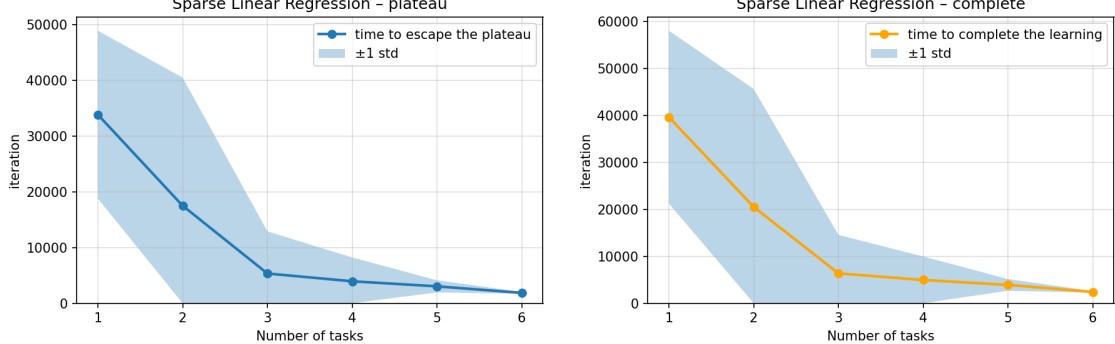

Figure 14: Sparse Linear Regression

# D Language ICL tasks

## D.1 Natural Language ICL tasks

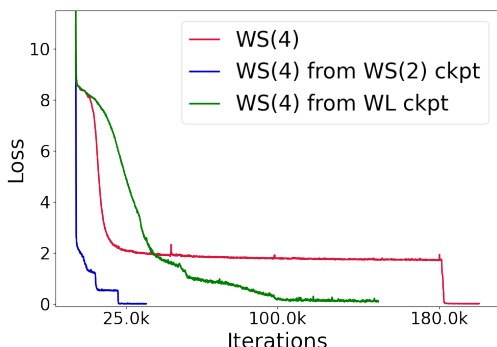

Figure 15: Checkpoint experiments on language ICL tasks.

To further verify our claim on tasks with real-world language data, we examine ICL in natural language processing (NLP) tasks.

**Model.** We use GPT2 (Radford et al., 2019) model with 12 layers, 12 attention heads, 768 embedding size, and 50257 vocabularies. We train GPT2 from scratch to learn in-context tasks.

**Dataset.** We use the word data from Nguyen et al. (2017). Following the approach in Todd et al. (2024), we choose 5245 words that can be tokenized to a single token. Words are then randomly chosen and arranged into sequences according to the task. Each sequence consists of six examples where the preceding five serve as context examples, and the last one as the query example.

**Prompting.** We convert a given sequence of words to a prompt by following Todd et al. (2024), where words are separated by a space, and examples are separated by a line:

```
<bos>Q. word_1^1 word_2^1 ... word_d^1
A. answer^1

⋮

Q. word_1^5 word_2^5 ... word_d^5
A. answer^5

Q. word_1^6 word_2^6 ... word_d^6
A.
```

**WordSelection**($d$)**.** We devise WordSelection($d$) by generalizing the WordSelection Task (Fu et al., 2024). The goal of this task is to select a single word from given $d$ words. The model's goal is to learn in-context which word to select.

**WordLength.** We also devise WordLength Task. Let $\text{len}(w)$ be the length of word $w$ and % be the remainder operator. Given four words $w_1, \ldots w_4$, the goal of this task is to learn in-context either $\sum_{i=1}^{4} \text{len}(w_i)$ or $\sum_{i=1}^{4} \text{len}(w_i)\%10$ from given examples.

**Result.** Prior work Fu et al. (2024) reported the difficulty of learning WordSelection(4), by showing that it cannot learn within about 78k iterations. Indeed, as shown in (Figure 7, Right), single-task training

of WordSelection(4) exhibits a plateau for over 180k iterations. However, mixing WordSelection(4) and WordSelection(2) significantly shortens the plateau to 3.2k iterations, accelerating the learning. The same phenomenon is observed when WordSelection(4) and WordLength are trained simultaneously. These results reveal that our claim generally holds in NLP tasks.

We also conduct the checkpoint experiment, where we use *checkpoint* models on WordSelection(2) and WordLength as baseline models and train them respectively on WordSelection(4). As shown in Figure 15, using *checkpoint* models shortens the plateau, demonstrating that NLP tasks also share a certain common structure.

### D.2   Formal Language ICL task: Regbench

We follow the configuration of Akyürek et al. (2024). To jointly train Quadratic Regression and DFA tasks, since their input formats differ, we use two different input embedding layers. Figure 16 illustrates the result.

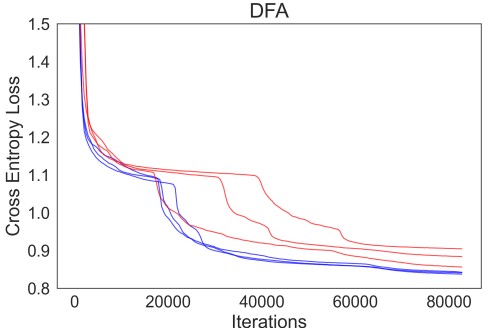
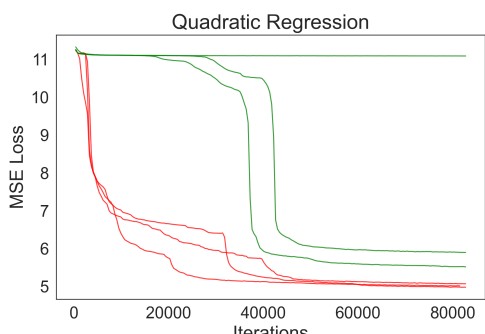

Figure 16: **Regbench.** DFA task shortens the Quadratic Regression's plateau, while Quadratic Regression does not shorten the DFA's plateau. Red line shows the multi-task training dynamics and blue and green lines show the single-task training dynamics.

# E   Checkpoint experiment

### E.1   Experimental details

In this section, we provide the configurations for the checkpoint experiments in Section 4.2.1. The input dimension is set to $d = 10$. For each single-task training, we save the model once it escapes the plateau. The exact iteration numbers where the models are saved are provided in Table 3. Since Sparse Parity(2) and Sparse Parity(3) do not escape the plateau within observable training steps, no checkpoint models could be obtained.

| | Boolean Task | Continuous Task | | | |
|---|---|---|---|---|---|
| | Boolean Retrieval | Linear Regression | Quadratic Regression | Sparse Linear Regression | Gaussian Retrieval |
| Iteration number | 105k | 19k | 12k | 7k | 50k |

Table 3: Checkpoint models.

After saving the baseline models, we train each task (Linear Regression, Quadratic Regression, Sparse Linear Regression, Sparse Parity(2), and Sparse Parity(3)) according to the configuration in Appendix A. To calculate the ratio, we divide the result by the plateau escape time summarized in the first row of Table 1.

**Learning time comparison of checkpoint models.** In Figure 6, we demonstrated that checkpoint models significantly shorten the plateaus. However, since the time required to escape the plateau and to learn the task may differ, it is necessary to verify whether the same phenomenon holds for learning time. Figure 17 confirms that this is indeed the case, although the effect is less robust compared to Figure 6.

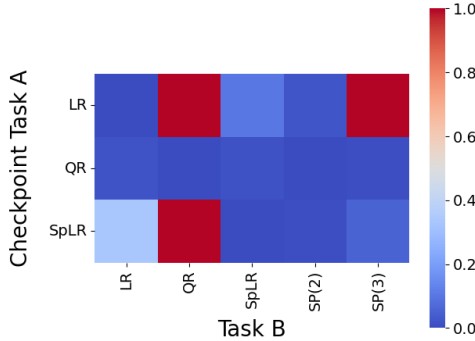

Figure 17: Learning time ratios of checkpoint models.

### E.2 Experiment on Retrieval task

Recall the description of Retrieval task.

**Retrieval.** Sample 1024 5-tuples of one key and four values: $\{(k_i, v_i^1, v_i^2, v_i^3, v_i^4)\}_{i=1}^{1024}$. The $k_i$s and $v_i^j$s are independently sampled from $\mathcal{D}_\mathcal{K}$ and $\mathcal{D}_\mathcal{V}$, respectively. To generate prompts, we uniformly sample $(n-1)$ 5-tuples without replacement and uniformly choose one $v_i$ per 5-tuple, resulting in $\{(\mathbf{k}_i, \mathbf{v}_i)\}_{i=1}^{n-1}$. Next, we sample $p \sim \text{Unif}(\{1, \ldots, n-1\})$ and set $\mathbf{q} = \mathbf{k}_p$. Finally, given $P_n = (\mathbf{k}_1, \mathbf{v}_1, \ldots, \mathbf{k}_{n-1}, \mathbf{v}_{n-1}, \mathbf{q})$, the task is to predict $\mathbf{v}_p$. We consider two Retrieval tasks: Gaussian Retrieval and Boolean Retrieval, with $(\mathcal{D}_\mathcal{K}, \mathcal{D}_\mathcal{V}) = (\mathcal{N}(0, I_d), \mathcal{N}(0, 1))$ and $(\mathcal{D}_\mathcal{K}, \mathcal{D}_\mathcal{V}) = (\text{Unif}\{\pm 1\}^d, \text{Unif}\{\pm 1\})$, respectively.

To train continuous ICL tasks, we use a checkpoint model saved from Gaussian Retrieval, and for boolean ICL tasks, we use a checkpoint model saved from Boolean Retrieval. We follow the configuration in Appendix A. For each experiment, we measure the time spent to learn each ICL task. The Results are illustrated in Figure 18.

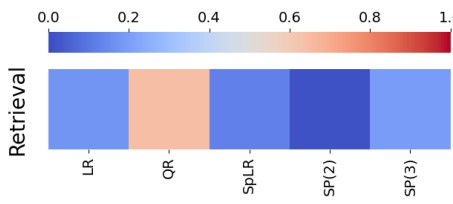

Figure 18: Checkpoint experiment on Retrieval.

## F  Feature learning experiment

The true function is $f_\star(x) = U\sigma(A_\star x)$, where $U \in \mathbb{R}^{k \times h}$ and $A_\star \in \mathbb{R}^{h \times d}$. Columns of $U$, denoted as $u_m$s, are independently sampled from $\mathcal{N}(\mu_m, I_d)$, with $(\mu_1, \ldots, \mu_k)$ being pre-determined vectors prior to training.

To design multi-task learning, we independently sample $k$ different vectors $\mu_1, \ldots, \mu_k$ from $\sqrt{h} \times \mathbb{S}^{h-1}$, making $\mathcal{N}(\mu_m, I_d)$s differ. In contrast, for single-task training, all $\mu_1, \ldots, \mu_k$ are set to the same vector $\mu$, which is also sampled from $\sqrt{h} \times \mathbb{S}^{h-1}$.

The model that learns the true function is $f(x) = V\sigma(Wx)$ where $V \in \mathbb{R}^{m \times h'}$ and $W \in \mathbb{R}^{h' \times d}$. We set $d = 150$, $h \in \{10, 20\}$, $h' \in \{100, 300\}$, and $k \in \{15, 30\}$. The Adam optimizer is used with a learning rate of $\text{lr} = 0.001$ and we apply $0.1 \times \text{lr}$ for the model's heads, following the approach in Berthier et al. (2024). We use the sigmoid activation function $\sigma(x) = \frac{1}{1+e^{-x}}$.

**Additional Figures.** Figure 7 represents the result when $(d, h, h', k) = (150, 10, 100, 15)$. To enhance the generality, we present figures for other hyperparameters as well. For each figure, multi-task training (blue lines) exhibits a much shorter plateau compared to single-task training (red lines).

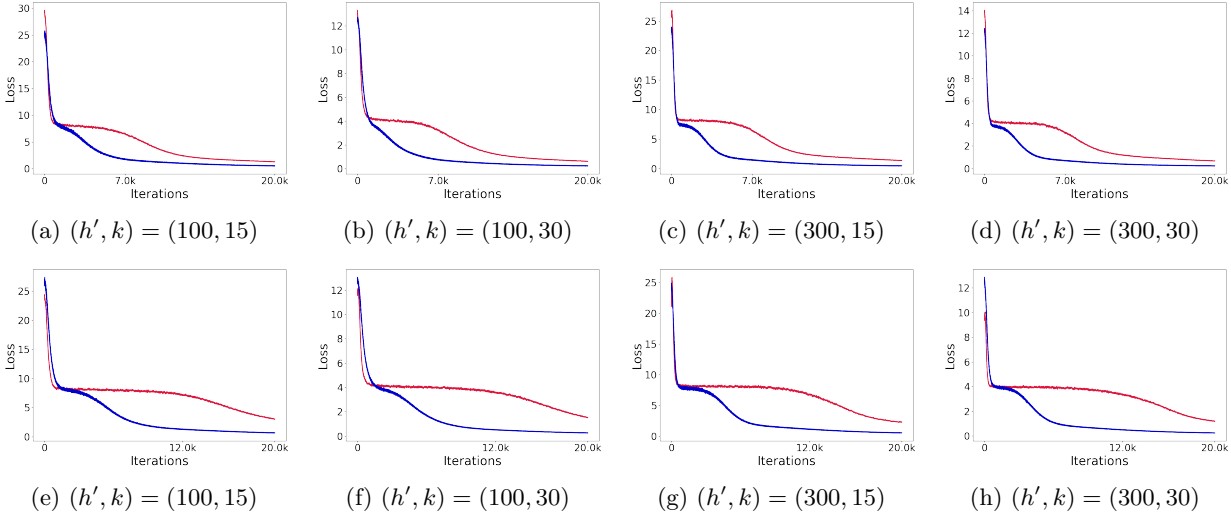

Figure 19: First row corresponds to the case when $(d, h) = (150, 10)$. Second row corresponds to the case when $(d, h) = (150, 20)$.

# G    No-context learning

## G.1    Optimal no-context functions

In this section, we provide a detailed explanation of the optimal no-context function. We have defined the optimal no-context function as follows. For continuous ICL tasks, $g^\star_{\mathcal{F}}: = \mathrm{argmin}_g \mathbb{E}_{f \sim \mathcal{D}_{\mathcal{F}}, x \sim \mathcal{D}_{\mathcal{X}}} \left[ (g(x) - f(x)^2) \right]$, where the argmin is taken over $\{g \mid g \colon \mathbb{R}^d \to \mathbb{R}\}$. For boolean ICL tasks, $g^\star_{\mathcal{F}}: = \mathrm{argmax}_g \mathbb{E}_{f \sim \mathcal{D}_{\mathcal{F}}, x \sim \mathcal{D}_{\mathcal{X}}} \left[ \mathbf{1}(g(x) = f(x)) \right]$, where the argmin is taken over $\{g \mid g \colon \{\pm 1\}^d \to \{\pm 1\}\}$.

For boolean ICL tasks, we can calculate $g^\star_{\mathcal{F}}(x)$ for each $x \in \{\pm 1\}^d$. The closed form of $g^\star_{\mathcal{F}}(x)$ can be expressed using $G_x: = \{f(x) \mid f \in \mathcal{F}\}$ as follows.

$$g^\star_{\mathcal{F}}(x) = \begin{cases} 1 & \#(1 \in G_x) > \#(-1 \in G_x) \\ -1 & (\text{otherwise}) \end{cases}.$$

For continuous ICL tasks, we cannot apply the same approach since the output space is not discrete. However, we can still derive the closed form for $g^\star_{\mathcal{F}}(x)$ for most tasks: $g^\star_{\mathcal{F}}(x): = \mathbb{E}_{f \sim \mathcal{D}_{\mathcal{F}}} [f(x)]$. This is because

$$g^\star_{\mathcal{F}}(x): = \mathrm{argmin}_{a \in \mathbb{R}} \mathbb{E}_{f \sim \mathcal{D}_{\mathcal{F}}} \left[ (f(x) - a)^2 \right] = \mathrm{argmin}_{a \in \mathbb{R}} \left( a - \mathbb{E}_{f \sim \mathcal{D}_{\mathcal{F}}} [f(x)] \right)^2.$$

For tasks other than LeakyReLU Regression, we can obtain the closed forms of $g^\star_{\mathcal{F}}(x)$ using the above property, as listed follows.

- Linear Regression: $g^\star_{\mathcal{F}}(x) = (\mu, \ldots, \mu)^\mathsf{T} x$.

- Quadratic Regression: $g^\star_{\mathcal{F}}(x) = x^\mathsf{T} U x, (U)_{ij} = \mu$.

- Sparse Linear Regression: $g^\star_{\mathcal{F}}(x) = \frac{k}{d}(\mu, \ldots, \mu)^\mathsf{T} x$.

- Decision Tree: $g^\star_{\mathcal{F}}(x) = \mu$.

For LeakyReLU Regression, we can estimate $g^\star_{\mathcal{F}}(x)$ with small error through a sufficient number of samples from $\mathcal{D}_{\mathcal{F}}$.

### G.2 Experimental details and additional figures

Let tasks with in-context function classes $\bigcup_{m=1}^{k} \mathcal{F}_m$ be trained simultaneously. Note that we do not discern the single-task and multi-task training: $k$ can be 1 as well. During training, we measure

$$\mathbb{E}_{f \sim \mathcal{D}_{\mathcal{F}_m}, x_1, \ldots, x_n \overset{\text{IID}}{\sim} \mathcal{D}_{\mathcal{X}_m}} |M_\theta(P_n) - g^\star_{\mathcal{F}_m}(x_n)|^2, \quad P_n = (x_1, f(x_1), \ldots, x_{n-1}, f(x_{n-1}), x_n)$$

for continuous ICL tasks and

$$\mathbb{E}_{f \sim \mathcal{D}_{\mathcal{F}_m}, x_1, \ldots, x_n \overset{\text{IID}}{\sim} \mathcal{D}_{\mathcal{X}_m}} \left[ \mathbf{1}(\text{sign}(M_\theta(P_n)) \neq g^\star_{\mathcal{F}_m}(x_n)) \right], \quad P_n = (x_1, f(x_1), \ldots, x_{n-1}, f(x_{n-1}), x_n).$$

for boolean ICL tasks. As shown by the following figures, these measurements are close to 0, which indicates that the model's output corresponds to *task-wise optimal no-context function*. We follow Appendix A for other configurations. Moreover, to prevent the optimal no-context be merely a zero function, we use $\mu \neq 0$ to parameterize the in-context function classes.

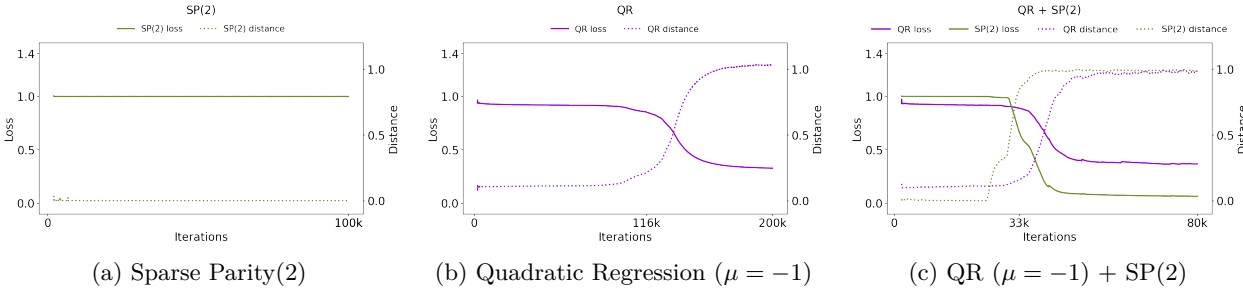

(a) Sparse Parity(2)      (b) Quadratic Regression ($\mu = -1$)      (c) QR ($\mu = -1$) + SP(2)

Figure 20: **No-context learning regime**. During the plateau, the model output matches the task-wise optimal no-context function.

