# OpenReview forum: "Task Diversity Shortens the In-Context Learning Plateau"
_TMLR — Accepted by TMLR_

### Review · Reviewer_wB6h · 2025-05-13

**Summary Of Contributions:**

This paper studies the observation that training a model on a diverse set of tasks/functions makes the model learn more quickly than training it on more data from a single task. The authors hypothesize that common structure shared among a set of functions/tasks is what allows it to learn this structure more easily, because that structure is present across multiple "views" from different tasks.

**Audience:**

Yes

**Broader Impact Concerns:**

None, although I'd suggest elaborating at least a bit on "potential societal consequences" as mentioned in the current statement.

**Claims And Evidence:**

Yes

**Requested Changes:**

- Please clarify the difference between "statistical" benefits of task diversity vs. "optimization" benefits.
- I'm confused why the test error is lower than training error in Figure 1.
- Please clarify how many runs are used for each training curve shown in the paper.
- Please specify why the particular tasks were chosen (beyond that they were used in prior work).
- In Table 2, please clarify which model was used in this experiment, and which task received 1/2 of the examples during training. Or, is each column a different training run? What is the difference between the number in and out of parentheses?
- Typo: "raining loss" in Figure 5
- Can you provide some analysis on the kinds of examples the model is learning to get correct in the Sparse Parity(2) task in Section 4.1 when its accuracy is roughly 55% for both train and test?
- I don't quite understand the claim made in Section 4.1 -- is it that in single-task learning, the model has learned the function itself, rather than learned to use the context?
- How is "escaping" the plateau identified?

I realize some of the details above (e.g., how plateau escape is identified) may be experiment-specific and present in the supplementary -- in this case, I'd suggest including a general description of the method (e.g., identifying a point in time when the loss/accuracy has reached some threshold) in the main paper.

**Strengths And Weaknesses:**

Strengths:
- The paper provides compelling evidence of and investigation into the benefits of task diversity for learning.

Weaknesses:
- Some of the terminology is rather vague, which makes the claims more difficult to understand. See Requested Changes for examples.

---

> ### Author Response · Authors · 2025-06-20
> **Author Rebuttal**
>
> We appreciate the reviewer’s positive evaluation and the insightful comments and questions. Below, we answer the reviewer’s questions.
>
> 1. Statistical benefits VS Optimization benefits
> - Statistical benefit is about generalization. With multitask training, the model sees more distinct samples, so each task requires fewer task-specific samples to reach a given accuracy. Optimization benefit is about training and the training loss. It relates to the loss landscape itself–mixing tasks alters the gradient field in a way that helps the model leave the flat ‘plateau’ region sooner.
> 2. Test error vs training loss in Figure 1
> - As noted on page 4, the test loss is measured only at the final prediction position of each context, evaluating whether the model has inferred the hidden function f. The training loss, by contrast, averages over all positions in the context. Since the test error does not take into account the intermediate tokens, it can be lower than the training loss.
> 3. Statistical confidence
> - Every experiment in Table 1 was repeated three times with different random seeds. The revised manuscript also includes standard deviation bands on all learning-curve plots (see page 18). These confirm that our claim is statistically robust.
> 4. Why we chose our ICL tasks
> - We adopted the synthetic ICL tasks most commonly used in prior work on ICL, but without any other reason.
> 5. Clarification on Table 2 experiments
> - We used GPT-2 model as in Table 1. For a detailed description of the model, please refer to Appendix A. The number inside the parentheses indicates the time until training is completed, while the number outside the parentheses represents the time it takes to escape the plateau. This notation is the same as in Table 1. In the first row (num_task = 1), tasks are not mixed; only the task corresponding to the column is used for training. In the second row (num_task = 5), tasks are mixed, and the task corresponding to the column receives half of the examples. For example, the value 2.4k (2.9k) in the first column of the second row means that SP(2) receives half of the examples, while the remaining 4 tasks each receive 1/8 of the examples. We agree that the explanation of Table 2 was insufficient. We will add more details to the appendix.
> 6. No-context learning
> - Yes, the no-context function depends only on the function class, not on any particular context. The linear regression, for which in-context functions are $f(x) = \langle w , x \rangle, w \sim N( \mu , I)$, the (optimal) no-context function is $g(x) = \langle \mu, x \rangle$. Here, $g$ is fixed by the class parameters. During the plateau, the model’s task is essentially to infer $\mu$ from the observed contexts.
> The same principle applies to Sparse Parity (2). Regardless of which subset A defines the in-context sequence, the no-context predictor outputs a constant $(1,-1)$ for any boolean vector. For example, during the plateau, the model will always predict $+1$ for $x = (1,1,-1)$, independent of the choice of $A$.
>
> 7. Definition of escapling plateau
> The plateau escape time t_plateau is defined as: $t_{plateau} = \min_{t > 100} [ (1/100) * \sum_{t' = t-99}^{t} \text{Training loss}(t') < 0.8 ]$ In other words, we consider the plateau to be escaped when the average training loss over 100 consecutive iterations falls below 0.8. This effectively captures the plateau escape, as we have normalized each task’s loss to be around 1. We had originally described this in Appendix A, but we will move this explanation to the main paper for clarity.

---

### Review · Reviewer_bBGm · 2025-05-29

**Summary Of Contributions:**

The paper studies the optimization behavior of models trained from scratch on in-context learning tasks. Prior work documents a loss plateau followed by rapid learning on single-task ICL; this paper considers the relative optimization behavior of single-task and multi-task ICL.

The authors show that training with two or more ICL tasks at once results in shorter loss plateaus than training on any individual task. They demonstrate that this behavior is partially but not fully transferable to a sequential training setup by taking a checkpoint trained on one task to the end of the plateau and training it on a second task; a loss plateau for the second task exists in this setting but is very brief. They hypothesize that the loss plateau is a period of _no-context learning_, where the model outputs the optimal "no-context" answer -- e.g. the single value with minimal test error --  for all inputs.

**Audience:**

Yes

**Claims And Evidence:**

Yes

**Requested Changes:**

I'd most like to see more analysis of the situations where task diversity does not shorten the learning plateau for one or both of the tasks; however, even without this, I think the paper makes an interesting contribution.

Other, more minor suggestions:
- slightly more careful language around the applicability of the claims to NLP/LLMs
- I think Appendix B could be a section of the main paper-- the results are quite interesting, and it seems you do have space to include it in the main body.
- typo in Figure 5 caption: "raining" -> "training"

**Strengths And Weaknesses:**

This is an interesting (and counterintuitive) finding, and I think it is both explained well and studied carefully. To the best of my knowledge, the concept of the loss plateau as representing a period where the model provides the optimal no-context answer is also novel, and both helpful intuitively and substantiated empirically. I also think the paper is strengthened by the inclusion of non-transformer architectures and by the brief study of language tasks in the appendix.

I think the claims about applicability to NLP in B.1 should be moderated slightly, as these tasks remain quite artificial and differ substantially from the complexities of tasks with real language data. Similarly, I think the claim in both the end of the abstract and end of the intro that this could be a factor in large-scale language model training should either be adjusted to indicate further uncertainty or substantiated with stronger evidence; I think this is an interesting phenomenon and it's certainly possible that it plays a role in language model pretraining, but there's really not a lot of evidence to draw this conclusion from studying much smaller models trained from scratch in limited data regimes.

I also think it would be interesting to include more study of tasks that have non-reciprocal loss plateau shortening-- e.g. quadratic regression and DFA in B.2-- or task pairs that don't exhibit this behavior at all, like training Boolean Retrieval before Boolean ICL. Can you characterize any properties of such tasks? The titular claim is that task diversity is helpful, but it seems this is not always the case-- is it possible that tasks that are diverse _to a point_ are useful (e.g. as you hypothesize, to learn formatting), but tasks that are higher diversity in form do not show optimization benefits from multi-task training. More characterization of the situations where this phenomenon occurs would strengthen the work.

---

> ### Author Response · Authors · 2025-06-20
> **Author Rebuttal**
>
> We appreciate the reviewer’s positive evaluation and the insightful comments and questions. Below, we answer the reviewer’s questions.
>
> 1. Features on non-reciprocal shortening cases
> - We don’t yet have a definitive answer because we have neither identified the common structure shared across ICL tasks. We believe a mechanistic explanation for our phenomenon would further elucidate, but we feel this is within the scope of future work.
>
> 2. Argument on NLP tasks
> - We agree that our current NLP experiments are not strong enough to prove that our finding fully transfers to LLMs. Still, our results on real-language ICL tasks indicate that the phenomenon likely holds for tasks formulated in natural language. We will moderate the relevant claims slightly, as suggested by the reviewer.

---

### Review · Reviewer_UhrC · 2025-06-05

**Summary Of Contributions:**

Training on a mixture of diverse in-context-learning tasks deal with the loss plateaus that plague single-task ICL.

The authors show this effect across six synthetic regression/parity tasks, two state-space models (Hyena, Mamba), a GPT-2-like transformer, and they demonstrate that a checkpoint that escapes the plateau on one task jump-starts learning on others.
Together these results position task diversity as a powerful implicit curriculum for faster ICL optimisation, while highlighting open questions about the latent structure being learned and whether the speed-up persists in modern, billion-parameter LLMs.

**Audience:**

Yes

**Claims And Evidence:**

No

**Requested Changes:**

please adress the weaknesses

**Strengths And Weaknesses:**

# Strenghs

Clear empirical result -  cutting optimisation time across six synthetic tasks and three very different architectures is very nice divercity in results.

Convincing transfer test - a checkpoint that has escaped the plateau on Task A instantly accelerates Task B, backing the claim that multitask exposure uncovers reusable structure rather than just adding data


# Weaknesses

The paper rules out induction heads via Retrieval checkpoints but offers no concrete mechanistic account.

The models are outdated, would the findings stay valid with more current LLMs, as well as reasoning models.

ICL papers turn to use real data for the minimum text classification (AGNews, sentiments and so on), the proxy experiment may not lead to real world findings and the authors should conduct this experiments.

Escape time is defined by a fixed training-loss threshold (0.8). Different tasks have different loss scales even after normalisation—robustness to threshold can be shown by experiments? e.g. it's not task agnostic and need manual tune?

Statistical confidence. Plots show single training runs; variance over random seeds / initialisations is not reported and known to have large impact in ICL due to the choise of examples. I would suggest to show mean+-std.

Batch-size confound. Multi-task batches contain more distinct functions per update. Could the speed-up stem from effective larger gradient variety rather than diversity per se?

Links to broader optimisation literature (e.g. curriculum learning, data-ordering effects) are only briefly mentioned. This is not a question but a kindly note to make a greater link to literature.

Normalising losses to plateau at 1 is convenient, but could mask dynamics; ablations without this heuristic are absent about the coefficients.

---

> ### Author Response · Authors · 2025-06-20
> **Author rebuttal**
>
> 1. Question regarding the induction head
> - Section 4.3’s retrieval-task experiment demonstrates that the shared structure among our ICL tasks goes beyond the induction head. Identifying the precise algorithm responsible is, at present, shown to be hard for non-simple setups of transformers [1]. Prior work has likewise shown that transformers deploy several overlapping mechanisms for ICL. We therefore limit our claim to showing that some richer, cross-task mechanism is at work.
> 2. Choice of plateau metric
> - Although we define ‘plateau escape’ as the first step where the normalized falls below 0.8, the drop from plateau to the learned regime is extremely sharp for every task (see Appendix B). Thresholds in the 0.3-0.8 range yield almost identical plateau escape points, so the qualitative conclusions are insensitive to this choice.
> 3. Statistical confidence
> - Every experiment in Table 1 was repeated three times with different random seeds. The revised manuscript also includes standard deviation bands on all learning-curve plots (see page 18). These confirm that our claim is statistically robust.
> 4. Batch-size confound
> - We are not yet convinced that “greater gradient variety” alone can explain plateau escape, but we would be happy to answer further once the reviewer clarifies the precise mechanism they have in mind.
> 5. Link to optimization literature
> - Thank you for this comment. We would be happy to expand the literature review section if provided with specific suggestions.
>
> [1] Do Pretrained Transformers Really Learn In-Context by Gradient Descent?, ICML, 2024.

---

> > ### Comment · Reviewer_UhrC · 2025-07-05
> >
> > **Follow-up on “Batch-size confound”**
> >
> > > *We are not yet convinced that “greater gradient variety” alone can explain plateau escape, but we would be happy to answer further once the reviewer clarifies the precise mechanism they have in mind.*
> >
> > Thanks for the invitation to clarify.  In the current **k-task** schedule, each SGD step averages gradients from *k* **independent functions** (one per task), while the single-task baseline averages just one.  This boosts the *within-mini-batch diversity* of targets, which can (a) lower gradient variance along plateau-preserving directions and (b) introduce new directions that help the optimizer escape flat regions.
> >
> > 1. **Effective “batch-of-functions” vs. tokens**
> >    Token-level batch size is matched, but the multi-task run sees *k* distinct input-output rules per update, whereas the single-task run sees just one.  This difference can change the signal-to-noise ratio of the gradients that matter for leaving the plateau.
> >
> > 2. **Control: multi-function single-task**
> >    Keep the *task* fixed yet present *k* different sampled functions per step: draw $\(f^{(1)},\dots,f^{(k)}\)$ i.i.d. from the same distribution, concatenate their prompts, and update once.  If this alone shortens plateau escape to the level of the multi-task run, the speed-up is attributable (at least partly) to “function diversity per update” rather than cross-task transfer.
> >
> > 3. **Control: single-function multi-task**
> >    Conversely, down-sample multi-task batches so that each update uses gradients from **one randomly chosen task** while keeping all other hyper-parameters unchanged.  If the plateau shortening largely vanishes, that again points to batch diversity—rather than shared representations—as the primary driver.
> >
> > I’m **not claiming** this is the whole story. Your checkpoint-transfer results do suggest genuine cross-task structure. However, without these controls it is hard to separate representational transfer from a purely optimization-level benefit of more varied gradients per step.
> >
> > Hope this clarifies the mechanism I had in mind and i'll be happy to discuss further

---

> > > ### Author Response · Authors · 2025-07-20
> > > **Author response**
> > >
> > > Thank you for your comment. We would like to clarify our experimental setup in more precise terms. The procedure is detailed in Algorithm 1 (page 16).
> > >
> > > We define the batch size as B (i.e., number of prompts=B) and all prompts has length n (i.e., sequence length=n). For each gradient computation step, we average over B prompts, each of length n.
> > >
> > > 1. Single-task setting:
> > > We draw B functions i.i.d. from a single task T, and construct B prompts of length n, each governed by one of those functions. These prompts are then combined into a batch for gradient computation. If we understood your point correctly, this setting already matches your proposed "multi-function single-task" control, where each SGD step uses B i.i.d. functions from the same task.
> > >
> > > 2. Multi-task setting:
> > > There are k distinct tasks, denoted as T_1, ....,T_k. From each task T_i, we draw B/k functions i.i.d., and each of these B/k functions is used to generate one prompt, resulting in B/k prompts per task. Then we combine all k × (B/k) = B prompts into the final batch. Therefore, our experimental setup is matched not only at the token level but also at the function level. Regarding your "single-function multi-task" suggestion, our setup ensures equivalent function diversity, and the observed plateau shortening persists, strongly implying that multi-task training's benefit arises from task diversity and shared structures (as further supported by our checkpoint experiments in Section 4.2.1).

---

### Comment · Reviewer_UhrC · 2025-06-04
**Review for Task Divercity ICL**

The paper’s core contribution is the empirical finding that training a mixture of diverse (ICL) function classes dramatically shortens the long “loss plateaus” reported in prior single-task studies (mostly accelerate where multi-task is reqired could be trivial, but the ckpt from one to another is a true novel).

## Qeustions for the authors:

The paper rules out induction heads via Retrieval checkpoints but offers no concrete mechanistic account. what is the common structure?

The models are a bit outdated, would the findings stay valid with more current LLMs (deepskeep / llama3 or any other opensource model published during or after 2023?). Same question for reasoning models.

Choice of plateau metric. Escape time is defined by a fixed training-loss threshold (0.8). Different tasks have different loss scales even after normalisation—robustness to threshold can be shown by experiments?

Statistical confidence. Plots show single training runs; variance over random seeds / initialisations is not reported and known to have large impact in ICL due to the choise of examples. I would suggest to show mean+-std.

Batch-size confound. Multi-task batches contain more distinct functions per update. Could the speed-up stem from effective larger gradient variety rather than diversity per se?

Links to broader optimisation literature (e.g. curriculum learning, data-ordering effects) are only briefly mentioned. This is not a question but a kindly note to make a greater link to literature.

Normalising losses to plateau at 1 is convenient, but could mask dynamics; ablations without this heuristic are absent about the coefficients.


Add probing / causal-tracing to show which layers/heads encode the shared structure and how it changes at plateau escape. Even partial results (e.g. attention pattern visualisations) strengthen C2–C3.

---

### Decision · Action_Editor_eRNS · 2025-07-22

**Recommendation:** Accept as is

**Additional Comments:**

Please make sure to address all clarify concerns raised by the reviewers that were not yet addressed in the updated manuscript as such concerns are likely to be shared by future readers of the work.

**Audience:**

Yes

**Audience Explanation:**

The ability of models to learn in-context, that is, to get input-output pairs as part of their input and learn the function that governs their relation is a topic of substantial interest in the machine learning community. This work sheds light on this and highlights the role of diversity in the input distribution in the emergence of this skill.

**Claims And Evidence:**

Yes

**Claims Explanation:**

The paper provides sufficient empirical evidence for a non-trivial claim that using multi-task training can accelerate the ability of models to learn in-context. This was agreed upon by all reviewers who stated the evidence is sufficient and the use of multiple architectures, tasks, and experimental settings supports the main claim of the paper.